# The Double-Edged Sword of ROS in Muscle Wasting and COPD: Insights from Aging-Related Sarcopenia

**DOI:** 10.3390/antiox13070882

**Published:** 2024-07-22

**Authors:** S. M. H. Chan, S. Selemidis, R. Vlahos

**Affiliations:** Centre for Respiratory Science and Health, School of Health and Biomedical Sciences, RMIT University, Melbourne, VIC 3001, Australia; stavros.selemidis@rmit.edu.au (S.S.); ross.vlahos@rmit.edu.au (R.V.)

**Keywords:** oxidative stress, antioxidants, COPD comorbidities, microRNAs, inflammaging

## Abstract

An elevation in reactive oxygen species (ROS) is widely accepted to be a key mechanism that drives chronic obstructive pulmonary disease (COPD) and its major co-morbidity, skeletal muscle wasting. However, it will be perhaps a surprise to many that an elevation in ROS in skeletal muscle is also a critical process for normal skeletal muscle function and in the adaptations to physical exercise. The key message here is that ROS are not solely detrimental. This duality of ROS suggests that the mere use of a broad-acting antioxidant is destined to fail in alleviating skeletal muscle wasting in COPD because it will also be influencing critical physiological ROS-dependent processes. Here, we take a close look at this duality of ROS in skeletal muscle physiology and pathophysiology pertaining to COPD and will aim to gain critical insights from other skeletal muscle wasting conditions due to aging such as sarcopenia.

An elevation in reactive oxygen species (ROS) is widely accepted to be a key mechanism that drives chronic obstructive pulmonary disease (COPD) and its major co-morbidity, skeletal muscle wasting. However, it will be perhaps a surprise to many that an elevation in ROS in skeletal muscle is also a critical process for normal skeletal muscle function and in the adaptations to physical exercise. Thus, the key message here is that ROS are not solely detrimental. This duality of ROS suggests that the mere use of a broad-acting antioxidant is destined to fail in alleviating skeletal muscle wasting in COPD because it will also be influencing critical physiological ROS-dependent processes. Here, we take a close look at this duality of ROS in skeletal muscle physiology and pathophysiology pertaining to COPD and reflect on/garner critical insights from other skeletal muscle wasting conditions due to aging such as sarcopenia. 

Elevated ROS levels are a hallmark of COPD due to chronic exposure to pollutants and cigarette smoke, as well as the chronic inflammatory state that activates immune cells like neutrophils and macrophages [1]. Increased levels of ROS are the strongest driving force for airway inflammation, fibrosis and emphysema [2]. In the quadriceps muscles of COPD patients, increased nuclear and mitochondrial ROS are linked to oxidative damage, driven by NADPH oxidase activation and mitochondrial dysfunction [1]. Cigarette smoking further exacerbates oxidative stress in skeletal muscle, promoting muscle wasting [1]. Despite the clear role of ROS in COPD muscle pathogenesis, clinical studies with antioxidants have generally failed to alleviate muscle wasting. This is because ROS are crucial signaling molecules in normal muscle function and exercise adaptations [3]. During muscle contraction, ROS regulate calcium signaling and excitation–contraction coupling, and they stimulate adaptive responses, enhancing muscle resilience and growth [4], suggesting a balance between beneficial signaling and harmful oxidative stress is essential. A recent meta-analysis showed that antioxidant supplementation often does not benefit muscle performance in athletes and can even hinder performance by interfering with mitochondrial biogenesis and vascular function [5]. Therefore, while managing oxidative stress is important, antioxidant supplementation must be carefully considered to avoid disrupting the delicate balance necessary for muscle health and adaptation.

To understand the balance of ROS in skeletal muscle, it is crucial to examine their generation and regulation. ROS in skeletal muscle are produced mainly through mitochondrial respiration, NOX activity and other metabolic processes [6]. Antioxidant defenses, both enzymatic (e.g., superoxide dismutase, catalase) and non-enzymatic (e.g., tocopherol, ascorbic acid), help manage ROS levels to prevent oxidative damage [6]. In COPD, this balance is disrupted by chronic inflammation, which increases ROS production and impairs antioxidant defenses [2], leading to muscle dysfunction and wasting [1,2]. ROS like superoxide anions exhibit bi-phasic properties, optimizing muscle contractility while contributing to muscle wasting. Superoxide is primarily generated by NOX2 in the plasma membrane, crucial for muscle contractility, and by the mitochondrial electron transport chain, although it is not directly involved in contractility [7]. NOX4, located in intracellular compartments, continuously produces hydrogen peroxide, which supports mitochondrial biogenesis and insulin sensitivity, essential for muscle health. NOX4-derived hydrogen peroxide also facilitates muscle contraction by oxidizing ryanodine receptors, regulating calcium release necessary for contraction, and it promotes the activation of Nrf2, enhancing cellular antioxidant defenses [8]. This spatial–temporal understanding of ROS generation and regulation highlights the complexity of their roles, emphasizing the need for a balanced approach in managing oxidative stress in skeletal muscle.

While moderate ROS levels are essential for muscle function and adaptation, excessive ROS can cause oxidative damage and impair muscle function (Figure 1). Antioxidant defenses, regulated by Nrf2, play a crucial role in maintaining this balance by activating genes encoding enzymes like superoxide dismutase and catalase [6,9]. In COPD patients, chronic inflammation and exposure to cigarette smoke overwhelm these defenses, leading to muscle dysfunction due to elevated oxidative stress and depleted antioxidants [10]. Interestingly, similar oxidative stress mechanisms are observed in aging, suggesting that insights from aging research could benefit COPD management. Aging is characterized by increased ROS production and reduced antioxidant defenses, leading to muscle deterioration and chronic inflammation [11]. Exercise can enhance Nrf2 activation and antioxidant responses, potentially improving muscle function in COPD patients [12]. However, individual variability in COPD severity and other factors may affect exercise efficacy.

The complexity of oxidative stress in COPD and aging necessitates targeted, site-specific interventions that enhance sub-cellular antioxidant defenses [13]. MicroRNA (miRNA) therapy holds significant promise, as specific miRNAs are pivotal regulators of antioxidant enzymes [14]. In both aging and COPD, miRNA dysregulation exacerbates oxidative stress and muscle dysfunction. For instance, miR-146a and miR-155 downregulate superoxide dismutase expression, while elevated miR-21 in aged tissues decreases SOD2 and catalase expression [15]. Figure 1 highlights the potential of miRNA in mitigating age-related muscle decline and COPD-associated skeletal muscle wasting by reinstating redox balance. In a recent human phase 1b trial [16], CDR132L a specific antisense oligonucleotide and a first-in-class miR-132 inhibitor, was found to be safe and well tolerated without apparent dose-limiting toxicity. In 28 patients with heart failure, CDR132L resulted in a dose-dependent sustained reduction in the plasma level of miR-132, significant QRS narrowing and improvements in biomarkers for cardiac fibrosis with an effective dose at ≥1 mg/kg [16]. This is undoubtedly encouraging; however, the application of miRNAs in clinical settings is still in its nascent stage. While preclinical studies and this human phase 1b trial show promise, several challenges need to be addressed. These include the identification of specific miRNA targets, the development of safe and effective delivery systems and understanding of off-target effects. Furthermore, the complexity of miRNA-mRNA interactions and the variability in patient responses add to the challenges in miRNA therapeutics [17].

Previous studies have predominantly focused on how miRNAs affect muscle mass, with less attention being given to how miRNAs might alter antioxidant defense and redox balance, which may be closer to the root of muscle wasting in COPD and aging. The regulation of non-coding RNAs, particularly miRNAs, is crucial in muscle atrophy caused by aging and COPD [18]. In patients with COPD, the upregulation of miR-1, which modulates mTORC1 signaling, has been shown to be essential for muscle maintenance [19]. Borja-Gonzalez et al. [20] demonstrated a role of miR-181a in age-related muscle loss, marking miRNAs as key epigenetic regulators. In patients with COPD, miR-1, miR-133 and miR-206 were identified as crucial regulators of muscle phenotype and adaptation, with aging and nutritional abnormalities being significant factors [21]. Meanwhile, circulating levels of miR-21 and miR-206 have been demonstrated to be biomarkers for sarcopenia in respiratory diseases, including COPD [22], which may be extended to the aging context. The feasibility of miRNAs as therapeutic targets and biomarkers in aging-related skeletal muscle loss was also emphasized in a comprehensive review by Jung et al. [23]. Collectively, these findings underscore miRNAs’ critical role in muscle health and disease, presenting promising avenues for therapeutic interventions in COPD and age-related muscle disorders. 

Overall, focusing on precise cellular locations where ROS damage is most detrimental and developing targeted miRNA-based therapies hold promise for treating muscle wasting in COPD and aging, offering a nuanced approach to managing oxidative stress and muscle dysfunction. However, the application of miRNA therapy is not without challenges. These include the specificity of targeting, as miRNAs can affect multiple genes, potentially leading to off-target effects. The delivery of miRNAs to specific tissues or cells is also a significant hurdle. Additionally, ensuring the stability of miRNAs in the body, understanding the complex miRNA-mRNA interactions and managing the variability in miRNA expression are substantial challenges. Potential side effects, as well as regulatory and ethical considerations, also need to be addressed. Despite these limitations, miRNA therapy remains a promising field. Further research is needed to fully elucidate the roles of miRNA, overcome these challenges and develop effective therapies.

## Figures and Tables

**Figure 1 antioxidants-13-00882-f001:**
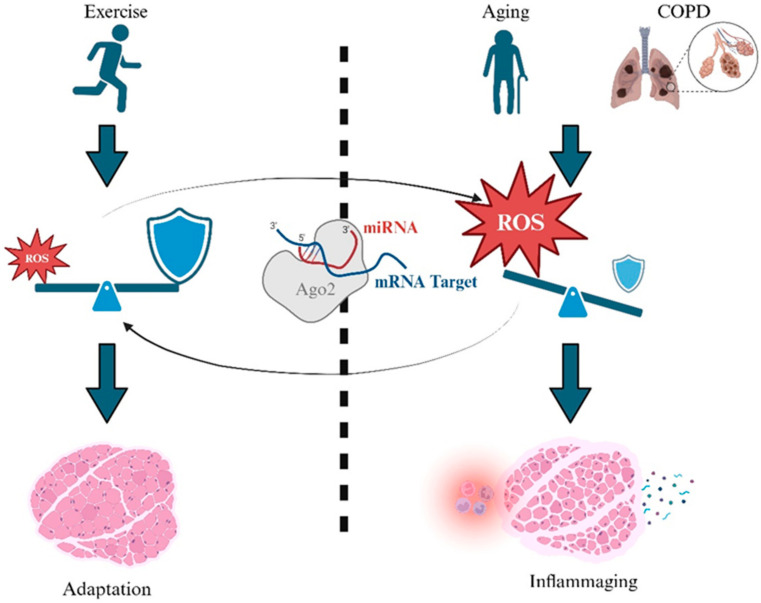
ROS as a double-edged sword for skeletal muscle adaptation and wasting, and the emerging roles of microRNA (miRNA) in regulating antioxidant capacity.

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
