# Peer review of "The Double-Edged Sword of ROS in Muscle Wasting and COPD: Insights from Aging-Related Sarcopenia"

_antioxidants, 2024, doi:10.3390/antiox13070882_

Round 1

Reviewer 1 Report

Comments and Suggestions for Authors

This commentary summarized the current understanding and authors’ opinions on the role of ROS in muscle wasting and COPD. It is well-written and logically formatted with sufficient supporting literature. A few minor suggestions are listed below.

1. Lines 71 to 75, the authors discussed the potential of miRNAs in improving muscle health in aging and COPD populations, which is also illustrated in Figure 1. Therefore, it would be better to include more details, such as the current stage of miRNA application and its limitations.

2. Line 21, is ---> are; Line 78, double-edge ---> double-edged.

Author Response

Comments 1: Lines 71 to 75, the authors discussed the potential of miRNAs in improving muscle health in aging and COPD populations, which is also illustrated in Figure 1. Therefore, it would be better to include more details, such as the current stage of miRNA application and its limitations.

Response 1: Thank you for pointing this out. We agree with this comment and have therefore added a significant description regarding the current stage of miRNA application in clinical settings and its limitation. The changes are marked in red in the "Marked" version of the revised manuscript page 3-5.

Comments 2: Line 21, is ---> are; Line 78, double-edge ---> double-edged.

Response 2: We have changed this in the revised manuscript, which can be found The change is marked in red in the "Marked" version of the revised manuscript page 6.

Reviewer 2 Report

Comments and Suggestions for Authors

In the current commentary, the authors reviewed the role of ROS in normal skeletal muscle physiology and muscle wasting in patients with COPD. It is a short but comprehensive overview of the role of ROS in skeletal muscle. The authors propose that in the future targeted miRNA-based therapies can be used to regulate pathological upregulation of ROS production/elimination in muscle.

Although it is a feasible approach, as with any other technique, it has limitations. I would suggest adding possible limitations of this technique that might be limiting factors in adapting this technique for use in clinical practice.

Author Response

Comments 1: Although it is a feasible approach, as with any other technique, it has limitations. I would suggest adding possible limitations of this technique that might be limiting factors in adapting this technique for use in clinical practice.

Response 1: Thank you for pointing this out. We agree with this comment and have therefore added a significant description regarding the current stage of miRNA application in clinical settings and its limitation. The changes are marked in red in the "Marked" version of the revised manuscript page 3-5.